# Effects of Light Exposure, Bottle Colour and Storage Temperature on the Quality of *Malvasia delle Lipari* Sweet Wine

**DOI:** 10.3390/foods10081881

**Published:** 2021-08-14

**Authors:** Elena Arena, Valeria Rizzo, Fabio Licciardello, Biagio Fallico, Giuseppe Muratore

**Affiliations:** 1Department of Agriculture, Food and Environment (Di3A), University of Catania, Via Santa Sofia, 100, 95123 Catania, Italy; elena.arena@unict.it (E.A.); biagio.fallico@unict.it (B.F.); g.muratore@unict.it (G.M.); 2Department of Life Sciences, University of Modena and Reggio Emilia, Via Amendola 2, 42122 Reggio Emilia, Italy; fabio.licciardello@unimore.it

**Keywords:** Malvasia, sweet wine, shelf-life, accelerated shelf-life test, 5-hydroxymethylfurfural, 2-furaldehyde

## Abstract

The influence of light exposure, bottle color and storage temperature on the quality parameters of *Malvasia delle Lipari* (MdL) sweet wine were investigated. Wine samples bottled in clear-colored (colorless, green and amber) glass were stored under different artificial lighting conditions, in order to simulate the retail environment (one cool-white, fluorescent lamp) and to perform an accelerated test (four and six cool-white, fluorescent lamps). The storage temperature was kept constant (25 °C) for the first 90 days of the experiment and then samples were monitored for up to 180 days at higher temperatures (30, 35 and 40 °C). The principal enological parameters, total phenols, color, 5-hydroxymethylfurfural (HMF) and 2-furaldehyde (2F) contents were studied. The shelf-life test pointed out minimum variations of the basic chemical parameters, while the quality attributes most affected by lighting were color, together with HMF and 2F levels which, hence, can be considered as indicators of the severity of storage conditions.

## 1. Introduction

The primary objective of packaging is to protect and retain, as much as possible, the quality of foods and beverages. The classical packaging material for wine is glass, appreciated primarily for its oxygen barrier [1], clarity and inertness: this feature, with respect to the migration of low molecular weight compounds from the package to the product and/or flavor scalping by the packaging material is of the utmost importance [2,3]. Wine has traditionally been stored in glass bottles of different colors and shapes, whose selection is often driven from market forces in the attempt to make the wine readily identifiable and more attractive to the consumer.

Wine shelf-life is defined as the time that it remains stable, in terms of its chemistry, microbiology and biochemistry as well as its sensory properties [4], however chemical changes may occur during storage, improving some quality parameters. Color is among the main sensory attributes determining consumers’ preference and it is considered a major feature for the assessment of red wine quality [5]. The color of white wines tends to brown after bottling [6], due to the effect of oxidation of the phenolic compound and to enzymatic reactions [7,8]. Thus, during storage and ageing, the chemical composition of wine is subjected to continuous changes, which may be desired or not, depending on the type of wine. Phenolics and volatile compounds as well as color and nutrients could be influenced by the lighting and temperature of the retail shops, and the optimization of packaging variables could contribute to wine quality preservation, from production to consumption.

Many authors have studied the effects of time, temperature and storage conditions on the phenolics composition and color of red wines [9,10,11]. A significant decrease of phenolics content and a change in the color of the white wine from pale yellow to yellow brown during storage, was reported. Time of storage seems to influence the color parameters and the total phenols content more with respect to temperature and light exposure [12]. Other authors [13] did not find color change, and total phenols decreased up to nine months of storage of Hellenic varietal white wine.

The deteriorative effect of light depends on other factors, such as the duration of light exposure, light spectra and intensity and the composition of the food product. It is well known that the exposure of some foods to ultraviolet radiation and visible light accelerates oxidative deterioration. In alcoholic beverages, the aroma, flavor and color easily deteriorate for light-induced oxidative processes. The effect of light can be explained by both photolytic autoxidation, which corresponds to the production of radicals during exposure to UV light, and photosensitized oxidation, that occurs in the presence of photosensitizers and ultraviolet or visible light. The roles of light, temperature and, more frequently, also the color of the glass bottle in white wine, have been extensively analyzed in order to evaluate their impact on light-induced oxidative degradation, changes on pigment, phenolic compounds, and mainly on enological parameters [14,15,16,17,18]. Clark et al. [19] discussed the specific determination of the critical wavelengths for photoactivation, leading to the formation of glyoxylic acid, considering the capacity of two types of glass bottles (clear and dark green) to protect or limit the photodegradation of tartaric acid to glyoxylic acid. It is well known that darker colored bottles tend to give greater protection to wine from the influences of light exposure on the assumption that dark colors do not allow the transmission of UV radiation. More recently, environmental groups have proposed that the wine industry should move away from dark green to flint or amber glass bottles as a contribution towards reducing recycling costs and energy demands [17].

The *Malvasia delle Lipari DOC* wine (MdL) produced in the Aeolian Islands (Italy) as a liqueur wine, with minimum developed alcoholic strength of 20% v/v, is one of the most ancient and aromatic wines of Sicily. The art. 6 of the Regulation D. P. R. 20/09/1973 ʺAssignment of the denomination of controlled origin of “Malvasia delle Lipari” wineʺ [20] describes the main characteristics of the MdL wine when it is released for consumption. The attention is also focused on the color, which must be golden yellow or amber yellow, underlining the importance of such a factor. Considering how the color could change during the storage, we focused on different colored glass bottles and different storage conditions during the shelf-life of the product.

According to production techniques, the grapes (Malvasia delle Lipari and Corinto Nero 95% and 5% respectively, as defined in D.P.R. 20/9/1973 [20]) are gathered when they are fully ripe and then sun-dried for 10–15 days on large mats made of bamboo canes, to reduce moisture and increase the sugar level (up to 32%) to obtain a much more aromatic white wine [21].

In MdL, the high level of sugars and the low pH value suggest the possibility to develop sugar degradation products, such as 5-hydroxymethylfurfural (HMF) and 2-furaldehyde (2F), generally considered as indicators of heat processing and/or the prolonged storage of foods [21].

Both HMF and 2F are the most important intermediate product of the acid-catalyzed degradation of hexose and pentose, respectively, and HMF derived also from the decomposition of 3-deoxyosone during the early stage of Maillard reaction [22]. HMF is used as a quality parameter in several processed foods [23,24,25,26,27,28,29]. In fortified wine with different sweetness levels, the concentrations of HMF were strongly dependent on time and temperature used during winemaking, and strictly related to the sugar content of the wine [30]. Moreover, both HMF and 2F are involved in the aroma of sweet, fortified white wines aged in oxygen-free conditions [31].

In recent years, several papers have debated on the safety of these compounds. HMF has been shown to be converted, in vitro and in vivo, into 5-sulfoxymethylfurfural, which has been reported to be cytotoxic, mutagenic and carcinogenic [32,33,34], and can be a poison for the nervous system [35].

The aim of this work was to study the influence of different lighting conditions, bottle color and storage temperature on the quality parameters of MdL wine: moreover, for the first time, the kinetics parameters, *k* and Ea, were determined for HMF and 2F formation in sweet wine.

## 2. Materials and Methods

### 2.1. Characterization of Coloured Wine Bottles

Clear glass bottles (200 mL) in three different colors (colorless, green and amber) were characterized in terms of light transmission properties, measuring on five glass samples exposed to a cool-white, fluorescent lamp (Osram Lumilux^®^ 36W/840, Munich, Germany) by a digital handle photo-radiometer (Delta OHM HD 2102.2, Delta OHM S.r.l., Caselle di Selvazzano (PD), Italy) equipped with probes for the measurement of illuminance (lux) (400–800 nm) and irradiance (W m^−2^) in the UVA (315–400 nm) and UVB (280–315 nm) regions. The light shielding effect offered by the packaging materials, expressed as percentage of irradiance, was calculated as:Light shielding= I_sample_/I_air_ × 100(1)
where I_sample_: irradiance recorded in the presence of sample; I_air_: irradiance recorded in the absence of sample [36].

### 2.2. Sampling

*Malvasia delle Lipari* DOC wine (MdL) produced in Sicily was kindly provided by a local manufacturer. The analytical parameters (pH, °Brix, total acidity, volatile acidity, alcoholic strength, total phenols, CIE Lab color parameters, HMF and 2F) were determined on the wine as received, before dispensing into the previously described clear-colored glass bottles (defined as Time “0”).

Bottles were fulfilled (200 mL with 7 mL headspace), then stoppered with crown corks and stored horizontally up to six months: three months under different lighting conditions and three months under different temperature conditions. Overall, 144 bottles were considered for the study of shelf-life, according to the experimental plan reported in Table 1.

During the first three months, the storage temperature was 25 °C. Samples were divided into three batches and exposed to different lighting conditions: (a) under constant illumination produced by one cool-white, fluorescent (CWF) lamp (Osram Lumilux^®^ 36W/840, Munich, Germany), as to simulate the retailers conditions; (b) under constant illumination produced by four CWF lamps and (c) under constant illumination produced by six CWF lamps, in order to perform an accelerated test, by exposing the sweet wines to extreme conditions. Lamps were placed 30 cm above samples. Table 2 reports the illuminance (lux) and UVA and UVB irradiance (W m^−2^) values recorded for each selected lighting condition.

After, for the next three months, MdL samples were stored under constant illumination, one cool-white, fluorescent (CWF) lamp, at three different temperatures: 30, 35 and 40 °C. Data loggers (Smart Reader SR04, ACR Systems Inc., Vancouver, BC, Canada) were used to monitor the storage temperatures throughout the trial.

MdL samples were periodically withdrawn at 30, 60, 90, 120, 150 and 180 days from being bottled and analyzed.

### 2.3. Analytical Parameters

Total acidity and volatile acidity, °Brix, pH and alcohol content were determined according to the OIV official methods [37]. Chromatic characteristics (CIE Lab) were determined according to Method OIV-MA-AS2-11 [37] using a spectrophotometer (Cary 1E, Varian, Leinì (TO) Italy) with a 1 cm quartz cell. The CIE parameters (*L**, *a**, *b**, C, h) were determined by the “Color Calculations” spectrophotometer software (Cary WinUV 1.3, Varian, Leinì (TO) Italy).

All analyses were carried out in duplicate, using chemicals (Sigma-Aldrich, Milan, Italy) and solvents of analytical grade (J. T. Baker, Deventer, The Netherlands) and HPLC grade (Merck, Milan, Italy).

### 2.4. HMF and 2-Furaldehyde

Aliquots of MdL sample were opportunely diluted with water (JT. Baker, Deventer, Holland), filtered through a 0.45-μm filter (Albet) and injected into an HPLC system (Shimadzu Class VP LC-10ADvp) equipped with a DAD (Shimadzu SPD-M10Avp, Kyoto, Japan). The column was a Gemini NX C18 (150 mm × 4.6 mm, 5 μm) (Phenomenex, Torrance, CA, USA), fitted with a guard cartridge packed with the same stationary phase. The HPLC conditions were the following: isocratic mobile phase, 90% water (Riedel-de Haën, Seelze, Germany) at 1% of acetic acid (Merck, Darmstadt, Germany) and 10% methanol (Scharlau, Sentmenat, Spain); flow rate, 0.7 mL/min; injection volume, 20 µL [26]. The wavelength range was 220–660 nm and the chromatograms were monitored at 285 nm. HMF and 2F were identified by comparison of the UV spectra and the retention time with those of HMF and 2F standard (*p* ≥ 99 % Sigma-Aldrich, St. Louis, MO, USA) and quantified using an external calibration curve. All analyses were performed in duplicate, including the sample dilution procedure, and the reported HMF and 2F concentration is therefore the average of four values.

Kinetic parameters for HMF and 2F formation in MdL stored under different lighting conditions and at different temperatures were calculated as reported by Arena et al. [25]. The activation energies Ea (kcal mol^−1^) values of HMF and 2F were calculated from rate coefficients at different temperatures by applying the Arrhenius equation.

### 2.5. Total Phenols

The total phenols (TP) analysis was performed as reported by Di Stefano et al. [38].

Briefly, 10 mL of wine was diluted 1:2 with 1 N H_2_SO_4_. Then, the diluted sample was passed through Sep Pack C18 cartridges (Waters Chromatography Europe BV, Etten-Leur, The Netherlands) previously activated with methanol and distilled water (2 and 5 mL, respectively). Adsorbed phenols were washed with 2 mL of 0.1 N H_2_SO_4_ and then desorbed with methanol (3 mL). The TP were assessed by the colorimetric Folin–Ciocalteu method at 700 and 760 nm, with gallic acid as calibration standard. Results were expressed as mg L^−1^.

### 2.6. Chemicals and Reagents

All reagents and solvents HPLC grade were purchased from Merck (Darmstadt, Germany). HMF, 2F and gallic acid standards were from Sigma (Milano, Italy).

### 2.7. Statistical Analysis

Statistical analyses were performed to assess the effect of packaging and storage conditions on the analytical parameters. The significance of differences was estimated by analysis of variance (ANOVA). The statistical significance level was set to 0.05. Statistical analyses were performed using SPSS^®^ Statistics 13.0 (IBM, Armonk, NY, USA).

Results are presented as mean value ± standard deviation. Data were analyzed through Spearman’s correlation to have a measure of the strength and direction of the association or relationship between the concentrations of analytical parameters and time of storage under different CWF lamp, and time of storage under different temperatures. All pairwise comparisons were run at 95% confidence intervals and p-values were Bonferroni adjusted through the statistical package SPSS^®^ Statistics 13.0 (IBM, Armonk, NY, USA).

## 3. Results and Discussion

### 3.1. Light Transmission Properties of Clear Glass Bottles

Figure 1 shows the light shielding expressed as percentage. As expected, the clear, colorless bottles offered a lower shielding effect in all spectra regions considered. In particular, shielding increased in the following order: colorless (10%), green (26.7%) and amber (49%) bottles for the illuminance parameter. A similar trend was observed for the UVA irradiance, while the shielding effect in the UVB region was 100% for both colored bottles and close to 90% for the clear bottle. This result is in agreement with results of Maury et al. [17], who compared the transmission spectrum of clear and green bottle colors and found that the former was capable of transmitting all visible and some UV light.

### 3.2. Effect of Light Exposure and Bottle Color on the Quality of Malvasia Delle Lipari Wine

MdL was characterized by the following base analytical parameters: pH = 3.79; °Brix = 1.35; total acidity= 5.4 g L^−1^; volatile acidity = 0.95 g L^−1^; alcohol content = 15.8% (*v*/*v*); TP = 247.10 mg L^−1^; HMF = 1.16 mg L^−1^; 2F = 1.11 mg L^−1^; *L** = 94.90; *a**= −2.99; *b** = 17.63; C = 17.88; h = 0.17.

Table 3 reports the changes in the quality parameters, as influenced by light exposure and glass bottle color over 90 days of storage.

Comparing the values of volatile acidity and pH at time “0”, both parameters showed the same course in all clear-colored glass bottles under all lighting conditions, with a slight decrease for volatile acidity up to 0.82 ± 0.02 g L^−1^. Total acidity ranged between 5.40 ± 0.34 g L^−1^ and 5.23 ± 0.06 g L^−1^ in all clear-colored glass bottles; samples bottled in clear colorless glass showed the same trend under 1, 4 and 6 CWF lamp lighting. Overall, small variations were observed during the shelf-life tests for pH, volatile and total acidity, but no statistical significance was observed (*p* > 0.05). Similarly, Revi et al. [18] as well as Hopfer et al. [39] found no differences (*p* > 0.05) in total and volatile acidity, pH and ethanol content of Chardonnay wine between bag-in-box containers and glass screw cap bottles after three months of storage at 20 °C.

Arapitsas et al. [40], studying white wine light-strike fault in flint and green glass bottles, reported any statistically significant difference among pH, titratable acidity and volatile acidity, and as the last one was not affected by time or packaging choice.

The trend for the TP during storage was the same for all clear-colored glass bottles and lighting conditions tested, reaching a mean value of 249.11 ± 18.66 mg L^−1^ after 90 days.

These results are discordant with previous observations where we noticed a reduction in TP during wine ageing, explained by the transformation and/or precipitation of phenolic material as oxidation reactions progress [41].

In MdL, this parameter did not change significantly, neither with different light exposure, nor with storage time (*p* > 0.05).

Color is one of the major attributes that affects the consumer perception of quality. Small changes in CIE parameters were observed. In particular, C, hue and *b** progressively increased at each lighting condition, from 0 to 90 days of the study. For instance, C changed during the storage time by a percentage of 11, 39 and 79% after 30, 60 and 90 days, respectively, in amber bottles under 4 CWF lamps. Similarly, the change ranged to about 7, 37 and 67% after 30, 60 and 90 days, respectively, in green bottles under 4 CWF lamps and up to 100% for colorless bottles under the same lighting. The *b** value increased proportionally through the storage period with the increasing number of CWF lamps, with R2 0.87, 0.90 and 0.95 after 30, 60 and 90 days, respectively. The increase in *b** values represents a higher intensity of yellow and the main increase is always in amber bottles, with only the exception of colorless bottles observed after 90 days and exposed under 4 CWF lamps. The *L** values, which give a value of the brightness of the samples, slightly decrease during 90 days of storage, until the 2–3% of reduction from the initial value for clear green and clear amber bottles under 6 and 1 CWF lamps respectively, while the decrease is near the 8% in colorless bottles exposed at 4 CWF lamps. The *a** value, which represents the degree of one component of green color, had a reduction during the first 30 days, and then an increase was observed after 60 days of storage under illumination, especially in clear amber bottles stored under 4 CWF lamps, according to Refsgaard et al. [42], while MdL stored in clear green bottles did not show any particular differences from the beginning. Hue (h), the attribute of appearance by which a color is identified according to its resemblance to red, green, yellow or blue, slightly decreases from 30 to 90 days. An increase in the yellow color (*b**) and hue is usually associated with wine aging, as well as a decrease in *a**, measured in light-exposed samples [43].

The HMF content showed a similar trend for the clear amber, green and colorless bottles, while some differences were observed when MdL was stored under different lighting conditions, mostly at 4 and 6 CWF lamps. In these cases, a rapid HMF increase was observed in the first 60 days, up to 18.56 ± 0.37 mg L^−1^, with a progressive increase over the following 30 days, up to 36.43 ± 0.97 mg L^−1^ (Table 3). Based on two-way ANOVA, the interaction effect of bottle color × lighting conditions on HMF concentration was not significant, while a significant effect (*p* < 0.001) was found for the lighting conditions, as expressed with different letters in Figure 2. It is well known that several factors influence the formation rate of this compound, such as temperature, time and storage conditions [26], as well as sugar concentration and water activity [44]. HMF reacted with ethanol to form 5-(ethoxymethyl) furfural, a compound founded in sweet fortified white wine, with a concentration above of the perception threshold, after 6 months of aging, in the absence of air [45].

As concerns 2F, produced by breakdown of pentose and/or Maillard reaction, from the starting value of 2F (1.11 mg L^−1^), an increase in the 2F concentration was observed in samples kept under 4 and 6 CWF lamps, until the final values were 3.3 ± 0.3 mg L^−1^ and 2.9 ± 0.2 mg L^−1^, respectively. Similarly to HMF, a statistically significant effect of lighting on 2F concentration was observed F (3.30 = 5.473, *p* = 0.007, partial *η*^2^ =0.477).

A two-way ANOVA was conducted to examine the effects of the exposure to a different number of CWF lamps, and the color of the glass bottles on each dependent variable examined (total and volatile acidity, TP, HMF, 2F, color parameter and pH). Residual analysis was performed to test for the assumptions of the two-way ANOVA. Outliers were assessed by inspection of a boxplot, then data were verified for the non-normality and homogeneity of variances. Normality was assessed using Shapiro–Wilk’s normality test for each cell of the design, and the homogeneity of variances was assessed by Levene’s test.

The interaction effect between CWF lamps and bottle color (Table 3) on volatile acidity, TP, Chroma, Hue and pH, was not statistically significant (*F*-test—degrees of freedom for the interaction term 6 in (6, 69)). Finding a non-statistically significant interaction does not mean that an interaction effect does not exist. Therefore, an analysis of the main effect for a kind of lighting (number of CWF lamp) was performed, which indicated that the main effect was not statistically significant (*p* > 0.05). All pairwise comparisons run where reported 95% confidence intervals and *p*-values are Bonferroni-adjusted.

At constant temperature, a Spearman’s rank-order correlation (2-tailed) was run to assess the relationship between HMF and 2F concentration, with storage time under 1, 4 and 6 CWF lamp. Preliminary analysis showed the relationship to be monotonic, as assessed by the visual inspection of a scatterplot. There was a strong positive correlation between HMF and the number of CWF lamps, *r_s_* = 0.846, *p* < 0.01 (Figure 2); as well as a positive correlation between 2F and storage time, *r_s_* = 0.429, *p* < 0.05.

Table 4 reports the kinetics parameters of both HMF and 2F formation in MdL wine stored under different lighting conditions. All systems fit a pseudo-first-order equation [46] well, and *k* values for HMF formation were always higher respect to those determined for 2F formation, regardless of bottle color and lighting conditions.

The MdL samples stored under 1 CWF lamp showed the lowest *k* values for HMF formation (about 0.017 days^−1^) independently from the bottle color. As the light exposure increases from 1 to 4 CWF lamps, the *k* value for HMF formation increases up to about 0.034 days^−1^ and was similar to those determined for MdL stored under 6 CWF lamps (about 0.031 days^−1^).

Similarly to HMF, 2F shows the lowest *k* value under 1 CWF lamp, while it increased by 2.6 fold under the lightening of 4 and 6 CWF lamp, independently of the bottle color.

### 3.3. Effect of Storage Temperature on Quality Changes of Malvasia Delle Lipari Wine

After the first 90 days, samples were stored up to 180 days under constant illumination (one CWF lamp) at three different temperatures: 30, 35 and 40 °C (Table 5).

The pH value was stable during the first 90 days of the study, showing a slight decrease, reaching 180 days, especially at 30 °C and 35 °C, reaching values of 3.31 and 3.29 respectively; in addition, volatile acidity showed only slight changes and limited standard deviation.

The TP content at the end of the study was practically coincident with the initial values, notwithstanding the oscillations observed during the trials.

The Chroma (C) is the measure of a stimulus judged relative to the white. Chroma, hue and *b** followed the same path. Initial C value was 35.10 and its value increased from 90 to 180 days until an average value of 46.00 ± 8.30 among the three temperatures. C values increased with storage temperature, in the order: 30, 35 and 40 °C. No significant change was observed for *L**. The *a** value was stable for the first 150 days to increase in samples at 35 °C and 40 °C. From 150 to 180 days, an increase was also monitored in MdL stored at 30 °C, exceeding the values of all the other samples. As previously reported, an increase in color parameters is linked to light-exposed samples [43].

The HMF levels changed considerably, mostly when MdL was stored at the highest temperature. At 30 °C, the HMF content increased from about 9 mg/kg up to 37 mg/kg at 180 days of storage. At 35 °C, the HMF content increased up to about 125 mg/kg at the end of storage. The effect of temperature on HMF formation, obviously, was more evident at 40 °C. At this storage, the HMF increases rapidly, even after 120 days, and at 180 days of storage, reached the highest levels, about 153 mg/kg.

During MdL storage at 30 and 35 °C, 2F increased slowly and important changes were only founded at 40 °C. At this temperature, after 180 days of storage, the 2F concentration was about 3 times higher than that determined at 30 °C.

Moreover, HMF levels were always higher than 2F. The first furanic compound was derived from glucose and fructose, while 2F comes from pentoses present in wine, in minor amounts. This trend was reported by other authors in sweet wine [30].

To underline changes related to different storage temperatures, a Spearman’s rank-order correlation was carried out to assess the relationship between color parameters, HMF and 2F concentration, with storage time under different temperatures. Preliminary analysis showed the relationship to be monotonic, as assessed by visual inspection of a scatterplot. There were strong positive correlations (*p* < 0.01) between: HMF and 2F, *r_s_* = 0.972; HMF and temperatures, *r_s_* = 0.798; 2F and temperatures, *r_s_* = 0.739 as reported in Table 6 (the correlation between C and all the color parameters was significant for *p* < 0.01).

Table 7 shows the kinetic parameters *k* (days^−1^*)*, and the activation energy (Ea), both for HMF and 2F formation, in MdL wine stored at different temperatures. Concerning HMF, the lowest *k* value was determined at 30 °C (0.0144 days^−1^), and the highest was found at 40 °C (0.0167 days^−1^).

The kinetics for 2F formation exhibited a *k* value lower than those of HMF, ranging from 0.0038 to 0.0088 days^−1^, at 30 and 40 °C, respectively.

Cutzach et al. [45] studied the formation of some compounds, including HMF and 2F, during the aging of sweet fortified wines at the temperature of 37 °C. Data reported in this work concerning HMF and 2F were used to determine *k* value, in order to compare our data with those reported in literature. Furthermore, in this case, *k* value was lowest for 2F formation and highest for HMF formation and kinetic parameters were very similar to our data (2F, *k* = 0.0099 days^−1^, *R^2^* = 0.987; HMF, *k* = 0.0136 days ^−1^, *R^2^* = 0.992), suggesting a similar mechanisms of reactions.

As concerns the activation energy (Table 7), the lowest value was determined for HMF (11.7 kJ mol^−1^), and the highest, about 5.7-fold higher than HMF, for 2F formation (66.4 kJ mol^−1^), suggesting that a small increase of temperature induced a fast increase of the formation of these compounds, mostly for HMF.

In recent years, much research has been focused on the influence of light exposure to wine composition, in particular, on white wine, but also on red wine [41,43]. The phenomenon of light-induced off-flavors in wine, called “light-struck taste” (LST) [40,46], together with bottle color, storage condition and temperature effect were studied to improve the quality maintenance of wine during its storage [40,43,47,48,49]. To the best of our knowledge, no one has studied light interactions and bottle’s color, or the temperature’s effect on HMF and 2F, during a prolonged storage of 90 and 180 days, respectively.

## 4. Conclusions

The study was conducted in order to assess the changes in some quality attributes of MdL wine as a function of clear-colored glass bottles, lighting and temperature. Overall, the quality parameters of MdL wine were not significantly altered with the exposure to different lighting conditions and under different storage conditions, however, some changes in the color parameters occurred in every tested condition. In contrast with common beliefs, the bottle color did not play a significant role in the quality maintenance of this sweet wine. The significant increase in the levels of HMF and 2F was mainly dependent on the intensity of lighting and on the storage temperature, as confirmed by the activation energy, without noticeable effects of the bottle color. Results suggest that retailers have a higher responsibility in the product quality maintenance compared to the company’s choices of bottle color, which, in the specific case of MdL wine, can be merely driven by marketing consideration. Finally, HMF and 2F levels can be considered as global quality indicators for MdL wine during storage, offering a tool for detecting unsuitable storage conditions.

In conclusion, the quality maintenance of wine can be achieved by controlling environmental factors, such as light exposure and temperature, during its storage guaranteeing profits for wine producers and satisfaction for consumers.

## Figures and Tables

**Figure 1 foods-10-01881-f001:**
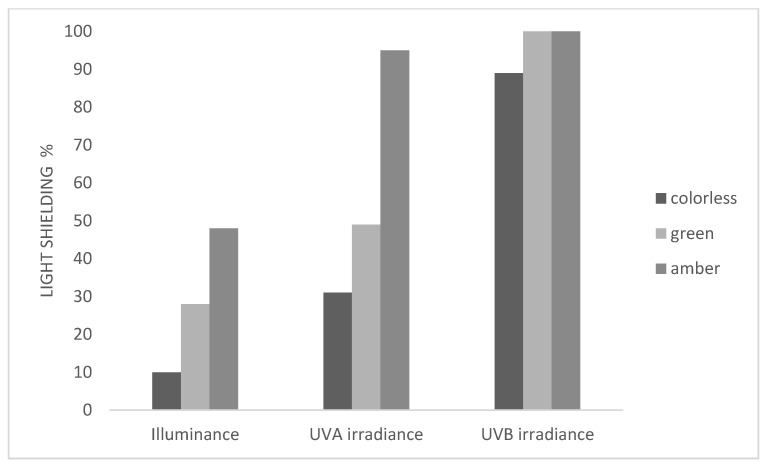
Light shielding (%) of clear glass bottles (colorless, green, amber).

**Figure 2 foods-10-01881-f002:**
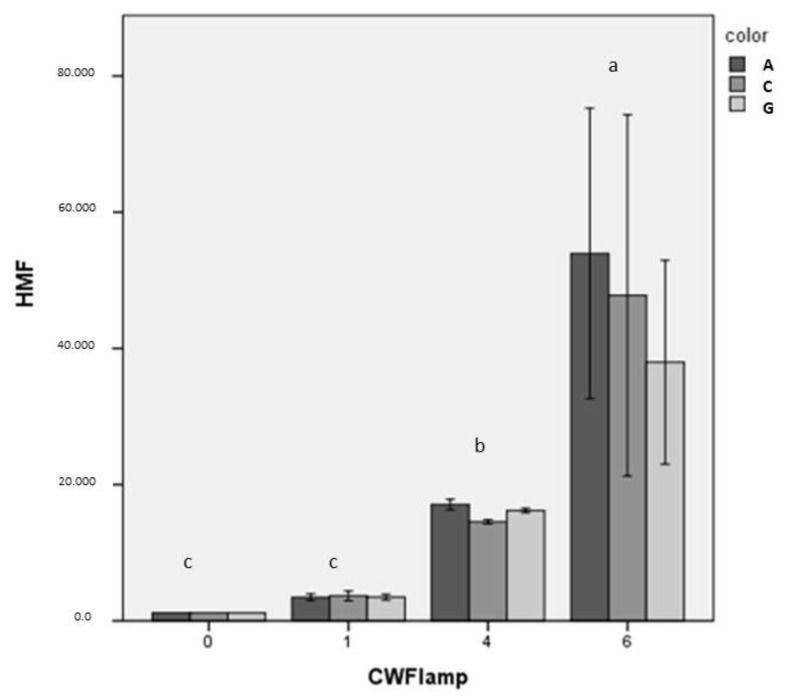
HMF concentration at constant temperature of 25 °C influenced by the number of CWF lamps (0, 1, 4, 6) and the color of clear glass bottles (amber, colorless, green); different letters represent significant differences (LSD test, *p* ≤ 0.05).

**Table 1 foods-10-01881-t001:** Experimental design.

Storage Conditions		Bottle Color	Storage Time
CWF Lamp	Temperature		
*n* = 4	*n* = 4	*n* = 3	*n* = 6
0 (control)	25 °C (control)	Colorless	30
1	30	Green	60
4	35	Amber	90
6	40		120
			150
			180

**Table 2 foods-10-01881-t002:** Measurements of illuminance (lux) and irradiance in the UVA and UVB regions (W m^−2^) recorded by digital handle photo-radiometer (values are mean of five measurements).

Storage Condition	Illuminance (lux)	UVA (W m^−2^)	UVB (W m^−2^)
1 CWF lamp	2671 ± 9	101.4^−3^ ± 0.1	10.71^−3^ ± 0.03
4 CWF lamps	9028 ± 16	284.5^−3^ ± 0.6	31.53^−3^ ± 0.06
6 CWF lamps	16127 ± 13	478.2^−3^ ± 0.8	53.91^−3^ ± 0.07

**Table 3 foods-10-01881-t003:** Quality parameters of MdL wine samples as influenced by light exposure (number of CWF lamps) and clear-colored glass bottles (A = amber; G = green; C = colorless) during 90 days’ storage (at constant temperature T = 25 ± 2 °C). (TP: total phenols; C: chroma; h: hue; *L**: lightness; *a**: redness; *b**: yellowness; HMF: 5-hydroxymethylfurfural; 2F: 2-furaldehyde).

Time (Days)	CWF Lamp	Bottle Color	TP (mg L^−1^)	C	h	*L**	*a**	*b**	HMF (mg L^−1^)	2F (mg L^−1^)
30	1	A	163.90 ± 3.40 ^mn^	18.76 ±0.3 ^de^	−0.11 ± 0.02 ^c^	94.22 ± 0.44 ^bc^	−2.14 ± 0.06 ^b^	18.63 ± 0.07 ^de^	2.08 ± 0.07 ^g^	1.02 ± 0.04 ^f^
		G	168.62 ± 0.88 ^l^	18.06 ± 1.1 ^e^	−0.09 ± 0.01 ^cd^	93.78 ± 0.55 ^bcd^	−1.62 ± 0.00 ^a^	17.99 ± 0.15 ^e^	2.23 ± 0.07 ^g^	1.24 ± 0.08 ^ef^
		C	158.31 ± 0.46 ^o^	17.41 ± 0.4 ^e^	−0.12 ± 0.01 ^bc^	94.60 ± 0.69 ^bc^	−2.01 ± 0.04 ^b^	17.29 ± 0.38 ^e^	1.89 ± 0.08 ^gh^	1.32 ± 0.09 ^e^
	4	A	179.40 ± 2.25 ^i^	19.97 ± 0.4 ^d^	−0.12 ± 0.2 ^bc^	93.69 ± 0.92 ^bcd^	−2.38 ± 0.08 ^bc^	19.82 ± 1.18 ^d^	4.62 ± 0.05 ^ef^	1.57 ± 0.09 ^d^
		G	160.94 ± 1.33 ^m^	19.06 ± 0.0 ^d^	−0.11 ± 0.01 ^c^	95.04 ± 2.56 ^b^	−2.00 ± 0.20 ^b^	18.96 ± 0.68 ^de^	3.51 ± 0.08 ^fg^	1.90 ± 0.09 ^c^
		C	144.06 ± 2.01 ^p^	18.79 ± 1.4 ^de^	−0.16 ± 0.3 ^ab^	94.79 ± 1.32 ^bc^	−3.00 ± 0.42 ^bc^	18.55 ± 1.66 ^de^	3.75 ± 0.07 ^fg^	1.50 ± 0.07 ^ed^
	6	A	193.26 ± 4.21 ^gh^	19.81 ± 2.0 ^d^	−0.11 ± 0.01 ^c^	93.23 ± 0.34 ^bcd^	−2.07 ± 0.01 ^b^	19.70 ± 0.49 ^d^	4.23 ± 0.05 ^f^	1.85 ± 0.09 ^c^
		G	190.31 ± 1.56 ^g^	19.50 ± 1.0 ^d^	−0.15 ± 0.1 ^b^	98.64 ± 6.29 ^a^	−2.85 ± 0.42 ^bc^	19.29 ± 0.36 ^d^	4.55 ± 0.08 ^f^	1.61 ± 0.07 ^d^
		C	192.51 ± 2.06 ^gh^	18.13 ± 0.2 ^de^	−0.16 ± 0.1 ^ab^	94.73 ± 0.51 ^bc^	−2.95 ± 0.39 ^bc^	17.89 ± 0.54 ^e^	3.98 ± 0.07 ^efg^	1.34 ± 0.09 ^e^
60	1	A	190.58 ± 0.94 ^g^	22.33 ± 0.1 ^c^	−0.17 ± 0.2 ^ab^	94.31 ± 0.02 ^bc^	−3.75 ± 0.01 ^cd^	22.01 ± 0.00 ^c^	4.26 ± 0.47 ^f^	1.06 ± 0.01 ^f^
		G	189.52 ± 1.10 ^g^	21.74 ± 0.1 ^cd^	−0.16 ± 0.01 ^ab^	94.23 ± 0.00 ^bc^	−3.41 ±0.00 ^c^	21.47 ± 0.01 ^cd^	4.28 ± 0.20 ^f^	1.62 ± 0.33 ^d^
		C	225.34 ± 2.95 ^c^	21.02 ± 2.3 ^cd^	−0.18 ± 0.0.14 ^a^	94.79 ± 0.01 ^bc^	−3.70 ± 0.01 ^cd^	20.70 ± 0.02 ^cd^	3.68 ± 0.12 ^fg^	1.70 ± 0.22 ^cd^
	4	A	205.22 ± 4.21 ^f^	24.83 ± 0.4 ^bc^	−0.20 ± 0.1 ^a^	95.29 ± 0.14 ^b^	−4.90 ± 0.02 ^de^	24.34 ± 0.01 ^bc^	18.56 ± 0.37 ^c^	2.03 ± 0.26 ^bc^
		G	210.42 ± 3.05 ^e^	24.61 ± 2.1 ^bc^	−0.18 ± 0.1 ^a^	94.54 ± 0.10 ^bc^	−4.25 ± 0.01 ^d^	24.24 ± 0.00 ^bc^	12.65 ± 2.24 ^cd^	2.65 ± 0.04 ^bc^
		C	220.26 ± 0.45 ^d^	22.83 ± 0.1 ^c^	−0.19 ± 0.02 ^a^	95.18 ± 0.10 ^b^	−4.31 ± 0.31 ^d^	22.42 ± 0.30 ^c^	14.01 ± 0.93 ^cd^	2.23 ± 0.04 ^bc^
	6	A	210.22 ± 2.01 ^e^	24.03 ± 0.4 ^bc^	−0.18 ± 0.1 ^a^	94.62 ± 0.01 ^bc^	−4.19 ± 0.00 ^d^	23.32 ± 0.02 ^bc^	11.37 ± 1.04 ^cd^	2.92 ± 0.53 ^ab^
		G	219.75 ± 1.32 ^de^	22.98 ± 0.2 ^c^	−0.18 ± 0.1 ^a^	94.70 ± 0.00 ^bc^	−4.15 ± 0.01 ^d^	22.70 ± 0.00 ^c^	10.44 ± 0.71 ^d^	2.32 ± 0.10 ^b^
		C	197.33 ± 0.47 ^fg^	25.25 ± 1.4 ^bc^	−0.19 ± 0.1 ^a^	95.11 ± 0.11 ^b^	−4.34 ± 0.00 ^d^	23.56 ± 0.01 ^bc^	10.80 ± 0.05 ^d^	1.83 ± 0.04 ^c^
90	1	A	236.10 ± 0.70 ^b^	29.59 ± 0.4 ^ab^	−0.11 ± 0.1 ^c^	91.89 ± 0.21 ^d^	−3.20 ± 0.23 ^cd^	29.42 ± 0.11 ^ab^	7.81 ± 1.21 ^e^	1.70 ± 0.09 ^cd^
		G	216.32 ± 1.74 ^de^	26.28 ± 0.1 ^b^	−0.15 ± 0.1 ^b^	93.49 ± 0.20 ^bcd^	−3.96 ± 0.12 ^cd^	25.98 ± 0.20 ^b^	9.92 ± 0.87 ^de^	1.81 ± 0.02 ^c^
		C	297.54 ± 0.40 ^a^	26.64 ± 0.2 ^b^	−0.14 ± 0.1 ^bc^	93.35 ± 0.20 ^bcd^	−3.76 ± 0.14 ^cd^	26.37 ± 0.41 ^b^	8.35 ± 0.47 ^de^	1.68 ± 0.18 ^cd^
	4	A	227.30 ± 3.74 ^c^	32.05 ± 0.4 ^a^	−0.16 ± 0.1 ^ab^	93.08 ±0.22 ^bcd^	−4.95 ±0.11 ^de^	31.66 ± 0.30 ^ab^	26.82 ± 0.87 ^b^	3.26 ± 0.05 ^a^
		G	228.84 ± 2.31 ^c^	29.94 ± 0.5 ^ab^	−0.15 ± 0.01 ^b^	93.01 ± 0.30 ^bcd^	−4.31 ± 0.10 ^d^	29.62 ± 0.21 ^ab^	27.35 ± 0.38 ^b^	2.97 ± 0.21 ^ab^
		C	236.14 ± 1.08 ^b^	35.94 ± 1.4 ^a^	−0.04 ± 0.01 ^d^	88.12 ± 0.43 ^de^	−1.60 ±0.00 ^a^	35.90 ± 0.20 ^a^	26.58 ± 0.85 ^b^	3.58 ± 0.15 ^a^
	6	A	224.31 ± 0.85 ^cd^	30.24 ± 2.1 ^ab^	−0.16 ± 0.1 ^ab^	93.39 ± 0.51 ^bcd^	−4.84 ± 0.31 ^de^	29.85 ± 1.10 ^ab^	36.43 ± 0.97 ^a^	3.00 ± 0.10 ^ab^
		G	220.57 ± 2.45 ^d^	28.03 ± 0.2 ^ab^	−0.15 ± 0.1 ^b^	92.78 ± 0.20 ^d^	−4.26 ± 0.30 ^d^	27.70 ± 1.00 ^b^	33.74 ± 0.10 ^ab^	2.82 ± 0.06 ^b^
		C	204.18 ± 1.04 ^f^	25.39 ± 0.4 ^bc^	−0.17 ± 0.1 ^ab^	94.26 ± 0.90 ^bc^	−4.35 ± 0.33 ^d^	25.01 ± 1.5 ^b^	30.75 ± 0.90 ^ab^	3.04 ± 0.18 ^ab^

Different letters within the same parameter and main factor show (bottle’s color) significant differences (LSD test, *p* ≤ 0.05).

**Table 4 foods-10-01881-t004:** Kinetics parameters for HMF and 2-furaldehyde formation in MdL stored in clear-colored glass bottles (amber, green, colorless) under different lighting condition (at constant temperature T = 25 ± 2 °C). (Control was a sample kept in the dark).

**5-Hydroxymethylfurfural**						
	**1 CWF Lamp**		**4 CWF Lamps**		**6 CWF Lamps**	
	***k* (min)**	***R*^2^**	***k* (min)**	***R*^2^**	***k* (min)**	***R*^2^**
Control	0.0116	0.91296	0.0116	0.91296	0.0116	0.91296
Amber	0.0164	0.95525	0.0363	0.98405	0.0316	0.99903
Green	0.0183	0.95012	0.0333	0.98975	0.0315	0.99805
Colorless	0.016	0.90128	0.0333	0.98967	0.0312	0.99953
**2-Furaldehyde**						
	**1 CWF Lamp**		**4 CWF Lamps**		**6 CWF Lamps**	
	***k* (min)**	***R*^2^**	***k* (min)**	***R*^2^**	***k* (min)**	***R*^2^**
Control	0.0044	0.7529	0.0044	0.7529	0.0044	0.7529
Amber	0.0048	0.9999	0.0115	0.9855	0.013	0.8765
Green	0.0056	0.967	0.0125	0.8969	0.0111	0.9769
Colorless	0.0054	0.8776	0.0125	0.9878	0.0101	0.9429

**Table 5 foods-10-01881-t005:** Evolution of quality parameters of MdL wine samples at different storage temperatures (30, 35, 40 °C) (HMF: 5-hydroxymethylfurfural; 2F: 2-furaldehyde; TP: total phenols; C: chroma; h: hue; *L**: lightness) from 90 to 180 days.

Time (Days)	T °C	HMF mg L^−1^	2F mg L^−1^	TP mg L^−1^	C	h	*L**
90	30	8.69 ± 1.10 ^h^	1.73 ± 0.07 ^e^	259.12 ± 1.62 ^bc^	31.95 ± 0.70 ^g^	−0.14 ± 0.00 ^a^	92.15 ± 0.27 ^a^
	35	26.92 ± 0.40 ^fgh^	3.27 ± 0.30 ^cde^	263.92 ± 9.23 ^bc^	37.61 ± 2.97 ^de^	−0.12 ± 0.00 ^bc^	91.13 ± 0.49 ^a^
	40	33.64 ± 2.84 ^fg^	2.95 ± 0.12 ^de^	250.02 ± 9.06 ^c^	35.75 ± 3.46 ^e^	−0.12 ±0.00 ^bc^	90.88 ± 0.86 ^ab^
120	30	17.60 ± 0.56 ^gh^	2.10 ±0.54 ^e^	257.62 ± 26.1 ^bc^	33.19 ± 0.15 ^ef^	−0.13 ± 0.01 ^ab^	91.98 ± 0.49 ^a^
	35	55.86 ± 2.19 ^de^	3.71 ± 0.33 ^cde^	249.62 ± 10.6 ^cd^	41.82 ± 0.41 ^c^	−0.10 ± 0.01 ^cd^	90.00 ± 0.39 ^ab^
	40	68.23 ±3.50 ^d^	4.71 ± 0.31 ^bcd^	284.77 ± 0.90 ^a^	38.47 ± 0.31 ^d^	−0.11 ± 0.01 ^c^	90.13 ± 1.33 ^ab^
150	30	29.70 ± 1.18 ^fg^	1.88 ± 0.19 ^e^	231.00 ± 4.53 ^ef^	34.62 ± 1.11 ^ef^	−0.13 ± 0.02 ^ab^	91.64 ± 0.48 ^a^
	35	101.43 ± 3.0 ^c^	5.24 ± 0.45 ^abc^	234.76 ± 30.48 ^e^	50.58 ± 1.94 ^ab^	−0.06 ± 0.02 ^e^	87.89 ± 1.13 ^bc^
	40	107.25 ± 9.7 ^bc^	5.07 ± 0.32 ^abcd^	246.72 ± 3.44 ^cd^	41.04 ± 4.32 ^c^	−0.09 ± 0.00 ^d^	89.86 ± 0.28 ^b^
	30	37.13 ± 1.86 ^ef^	2.02 ± 0.31 ^e^	238.42 ± 10.37 ^d^	40.76 ± 7.42 ^cd^	−0.06 ± 0.11 ^e^	87.24 ± 8.10 ^bc^
180	35	124.49 ± 15.54 ^b^	7.17 ± 1.76 ^a^	269.92 ± 16.60 ^ab^	55.57 ± 5.52 ^a^	0.05 ± 0.12 ^e^	76.26 ± 15.5 ^d^
	40	152.81 ± 12.18 ^a^	6.32 ± 1.69 ^ab^	239.00 ± 6.00 ^d^	41.67 ± 4.92 ^c^	−0.10 ± 0.02 ^cd^	89.60 ± 1.34 ^b^

Different letters within the same parameter and main factor show significant differences (LSD test, *p* ≤ 0.05).

**Table 6 foods-10-01881-t006:** Spearman’s rho T = 30-35-40 °C from 90 to 180 days of storage (*N* = 12). (HMF: 5-hydroxymethylfurfural; 2F: 2-furaldehyde: C: chroma; h: hue; *L**: lightness; *a**: redness; *b**: yellowness).

		HMF	2-Furaldehyde	C	h	*L**	*a**	*b**
**Time (days)**	Correlation coefficient	0.518 *	0.497	0.626 *	0.718 **	−0.777 *	0.791 **	0.626 *
	Sig. (2-tailed)	0.084	0.101	0.029	0.009	0.003	0.002	0.029
**T °C**	Correlation coefficient	0.798 **	0.739 **	0.414	0.313	−0.0325	0.312	0.414
	Sig. (2-tailed)	0.002	0.006	0.181	0.322	0.302	0.324	0.181
**HMF**	Correlation coefficient		0.972 **					
	Sig. (2-tailed)		0.000					
**C**	Correlation coefficient				0.919 **	−0.902 **	0.804 **	1.000 **
	Sig. (2-tailed)			.	0.000	0.000	0.002	
**h**	Correlation coefficient					−0.982 **	0.942 **	0.919 **
	Sig. (2-tailed)					0.000	0.000	0.000
***L** **	Correlation coefficient						−0.951 **	−0.902 **
	Sig. (2-tailed)						0.000	0.000
***a** **	Correlation coefficient							−0.804 **
	Sig. (2-tailed)							0.002

** Correlation is significant at the 0.01 level (2-tailed); * Correlation is significant at the 0.05 level (2-tailed).

**Table 7 foods-10-01881-t007:** Kinetic parameters for 5-hydroxymethylfurfural (HMF) and 2-furaldehyde (2F) formation in MdL stored under different temperatures (30, 35, 40 °C).

HMF				2F			
T (°C)	*k* days^−1^	*R* ^2^	Ea kJ mol^−1^ (kcal mol^−1^)	T (°C)	*k* days^−1^	*R* ^2^	Ea kJ mol^−1^ (kcal mol^−1^)
30 °C	0.0144	0.9538	11.7 (2.8)	30 °C	0.0038	0.6275	66.4 (15.9)
35 °C	0.0150	0.9448		35 °C	0.0070	0.9010	
40 °C	0.0167	0.9872		40 °C	0.0088	0.9488	

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
