# Peer review of "Effects of Light Exposure, Bottle Colour and Storage Temperature on the Quality of Malvasia delle Lipari Sweet Wine"

_foods, 2021, doi:10.3390/foods10081881_

Round 1
Reviewer 1 Report
The work is interesting. The methodical part is clear and logical.
The experiment was logically planned and implemented.
The results of research are presented readable. Tables and Figures clear .
In my opinion there is a lack of better discussion. The obtained results should be confronted with results of others researchers. The discussion part presented in the manuscript is insufficient.
Also more current references should be used in the manuscript.
Author Response
Dear Revisor
Thank you for the comments and suggestion to our submitted paper.
The comments reflect some considerations that we appreciated and the suggestion of improve the manuscript.
Thus, changes have been made according to the suggestions.
In particular new comments were added in Results and discussion paragraph, seven new and more recent papers (7) were cited and used to improve discussion on our results.

Reviewer 2 Report
The manuscript of Arena, Rizzo, Licciardello, Fallico and Muratore examines the influence of the storage conditions (bottle’s colour, temperature and lightning) over time on some physicochemical characteristics of an Italian DOC sweet wine - Malvasia delle Lipari.
The work is innovative and of interest for the scientific community and for the industry. The experiment was well designed. However, details and corrections in Materials and Methods section are needed to enable the reproducibility of the work. The results seem interesting but additional support based on the statistical analysis outcomes is required to attest their robustness. Major revision of the manuscript is needed based on the remarks.
- English language: authors should revise the manuscript taking into account that a unique English style should be used (UK or USA); in addition: L 12 (“was” instead of “were”); L 30 (“utmost” instead of “main”); L 49 (“change” instead of “alteration”); L 65 (“anecdotal evidence” is not an adequate expression, please replace by a suitable one); L 185 (“of” instead of “by”); L 285-292 (“regardless of” instead of “independently from”); L 326 (“carried out” instead of “run”); L 342 (“found” instead of “founded”); L 343 “lower than” instead of “lowest respect to”).
- Typing errors: L 282, 294 (kinetics); L 196-212 (L-1).
- Please replace “phenols” and “polyphenols” by “phenolics”.
- Title: please remove “changes”.
- L 34-35: the concept of shelf life is not directly applicable to wine because some chemical changes occurring during the storage in bottle may contribute to improve its quality, as mentioned in L 39-41. Please rephrase.
- L 55-58, 77-80: references are needed.
- L 70-72: despite being a DOC sweet wine of great importance for this region, why did the authors studied it? Is there any problem during its storage in the bottle? An explanation should be included in the text.
- L 71: please replace “alcohol levels of 20 degrees” by “alcoholic strength of 20% v/v”.
- L 99: what is the meaning of “clear”? Is it “colourless”? Please specify; check throughout the manuscript.
- L 101: (Astm, 2000) is not in accordance with the journal rules and is missing in the references’ list.
- L 98-107: without replicates? Without statistical analysis? How do the authors assure the reliability of results?
- L 115, 140: “alcoholic strength” instead of “alcohol content”.
- L 121: it is not clear the relationship between n=4 for the storage conditions because the experiment included three lightning modalities and three temperature modalities in each storage period. Please clarify. In addition, please include a scheme to facilitate the reader’s understanding.
- L 135-136: how was the temperature measured? Please specify.
- L 138-139: a subsection “Chemicals and reagents” or similar should be added to include all the chemicals used in the analyses made (subsections 2.3, 2.4 and 2.5).
- L 139: “parameters” instead of “indexes” because not all the results obtained were indexes.
- L 154: please provide information about the solvents, elution programme and injection volume used.
- L 158: how was the quantification made? Was it through calibration curves? How were the results expressed?
- L 170: How were the results expressed (mg/L or mg/Kg)? Please check the units throughout the manuscript.
- L 183: this sub-title should be more specific.
- L 201: “at the beginning”. Please specify.
- L 201-274, 298-324: the outcomes of means comparison test are required to assure the robustness of results and to support their discussion; “increase” and “”decrease” mean nothing without statistical analysis. Table 2, Table 4 and Figure 2 should include letters reflecting the outcomes of means comparison test.
- L 206: what is the meaning of “shelf life tests”? They were not described in Materials and methods section.
- Table 3 title: “MdL” instead of “Malvasia ...”; please decode “PET”.
- Table 3 headers: “neon”? Were not “CWF”?
- L 339-355: this sentences should be placed before Table 6.
- L 358: please remove “stresses”.
Author Response
Dear Revisor
Thank you for the comments and suggestion to our submitted paper.
Specific changes according to suggestions and comments are given in the attached file. Comments from editor are presented in dark letters, followed by our reply (red)
